# Assessment of SDG 3 Research Priorities and COVID-19 Recovery Pathways: A Case Study from University of the Western Cape, South Africa

**DOI:** 10.3390/ijerph22071057

**Published:** 2025-07-01

**Authors:** Josè M. Frantz, Pearl Erasmus, Lumka Magidigidi-Mathiso

**Affiliations:** 1Department of Physiotherapy, Faculty of Community and Health Sciences, University of the Western Cape, Bellville 7535, South Africa; 2Centre for Interdisciplinary Studies of Children, Families and Society, Faculty of Community and Health Sciences, University of the Western Cape, Bellville 7535, South Africalmagidigidi@uwc.ac.za (L.M.-M.)

**Keywords:** COVID-19, determinants, health priorities, interventions, pandemic recovery, research, Sustainable Development Goals, university

## Abstract

The COVID-19 pandemic has disrupted the progress toward Sustainable Development Goal 3, particularly in developing countries, exacerbating existing health disparities and creating new challenges for health systems worldwide. This study explores the role of university research in advancing SDG 3 targets in a post-pandemic context using the University of the Western Cape as a case study. Through qualitative data analysis of research titles and abstracts registered between 2020 and 2022, we applied the WHERETO model of McTighe and Bloom’s Taxonomy to categorize research according to the SDG 3 targets and indicators. This approach provides insight into which health priorities were addressed through scholarly research at UWC in alignment with the UN 2030 Agenda, particularly during pandemic recovery. Our findings indicate that research priorities largely corresponded with South Africa’s health challenges, with the highest concentration of studies addressing non-communicable diseases and mental health (Target 3.4), infectious diseases (Target 3.3), and medicine development (Target 3.b). These priorities align with the National Health Research Committee’s identified health priorities for disadvantaged communities in the Western Cape. Notably, research on mental health and emergency preparedness (Target 3.d) increased significantly during the pandemic period, reflecting shifting priorities in response to COVID-19. This study offers critical insights into how university research shifted priorities adapted during the pandemic and identifies areas requiring focused attention to support post-pandemic recovery. By highlighting research gaps and opportunities, our findings provide a foundation for developing more comprehensive approaches to health research that address the disparities exacerbated by COVID-19 while advancing the 2030 agenda. This model could inform research prioritization at other institutions facing similar challenges in both local and global contexts.

## 1. Introduction

The 2030 Agenda for Sustainable Development established Sustainable Development Goal 3 (SDG 3) with the mandate of “ensuring healthy lives and promoting well-being for all at all ages” [1]. Despite ambitious targets, progress toward SDG 3 has been slower than anticipated, with persistent maternal mortality ratios, infectious disease burden in lower-income countries, and significant disparities in accessing essential health services. These challenges have been further exacerbated by the COVID-19 pandemic, which has disrupted health systems worldwide and threatened to reverse decades of progress [2,3].

Universities occupy a strategic position in advancing the SDG agenda through their research capacity and ability to generate evidence-based solutions. As centers of knowledge production and innovation, universities can mobilize interdisciplinary expertise to address complex health challenges and inform policy decisions [4]. Through rigorous inquiry across medical, public health, and social science disciplines, universities can produce actionable evidence to prevent disease, strengthen healthcare systems, address social determinants influencing health, and raise global health standards [5]. Their interdisciplinary capabilities, when focused on translating research into practice, can advance SDG 3 priorities ranging from universal health coverage to controlling epidemics [6].

In South Africa, university research takes on particular significance against a backdrop of persistent health inequities. Despite the end of apartheid, stark disparities in health outcomes remain between racial groups [7]. South Africa continues to face a quadruple burden of disease: communicable illnesses like HIV/AIDS and tuberculosis, non-communicable diseases, maternal and child mortality, and violence-related injuries [8,9].

The COVID-19 pandemic has severely compounded these challenges, creating what scholars have termed “a pandemic within existing pandemics” in the South African context. When COVID-19 emerged in early 2020, it functionally added a fifth burden to an already strained health system, disrupting essential health services and exacerbating existing inequalities. This additional pressure has raised complex questions about balancing limited health budgets across competing priorities in health service provision and policy.

Recent analyses of post-pandemic health systems indicate that research institutions must now balance immediate recovery needs with long-term sustainable development objectives, requiring new frameworks for research prioritization [1]. South African universities, including the University of the Western Cape (UWC), have unique opportunities to inform provincial and national responses to these challenges through context-specific health research. Located in the Western Cape province and guided by a vision to become “an outstanding teaching and research university, driving positive change through science and knowledge” [10], UWC is well-positioned to mobilize research expertise addressing regional health priorities linked to SDG 3.

The university’s established research programs targeting HIV/AIDS, tuberculosis, non-communicable diseases, mental health, and injury prevention can be leveraged to support both pandemic recovery and longer-term sustainable development objectives. Additional networking with provincial Department of Health partners would further enable the translation of UWC research findings into improved programming and interventions.

The research landscape has been significantly altered by the COVID-19 pandemic, which disrupted conventional research activities while simultaneously creating urgent new priorities. The pandemic’s impact on research agendas provides a unique opportunity to examine how academic institutions respond to global health crises while maintaining a focus on longer-term sustainable development objectives. Understanding these dynamics is essential for developing a stronger alignment between university research and public health needs in post-pandemic recovery planning.

This paper aims to explore scholarly research conducted at the University of the Western Cape between 2020 and 2022 to assess how it addresses SDG 3 targets and to identify gaps requiring further attention. By analyzing research titles registered during this pivotal period, we seek to understand how pandemic conditions influenced research priorities and the university’s contribution to sustainable development in health.

As Sipido and Nagyova [11] note, translating research into SDG attainment requires adaptive strategies, comprehensive stakeholder engagement, and accelerated implementation approaches. Through this case study of UWC, we hope to provide insights that may inform similar efforts at other institutions in developing countries facing comparable health challenges and pandemic recovery needs.

The introduction of the SDGs, particularly SDG 3, represents a critical juncture for universities to direct their research agendas toward improving population health locally and globally [5]. As an anchor institution in the Western Cape, UWC has made substantial strides in addressing provincial health needs through context-specific research initiatives. However, realizing the full potential of university research for sustainable development requires stronger leadership, strategic priority-setting with government partners, and accelerated translation mechanisms. By seizing opportunities to promote comprehensive health research, universities can help save lives, reduce inequalities, and improve quality of life, essential objectives of SDG 3

## 2. Materials and Methods

### 2.1. Study Design and Analytical Framework

This study employed a qualitative case study approach utilizing document analysis as the primary research method. Document analysis involves a systematic procedure for evaluating documents, both electronic and printed, that requires the examination and interpretation of data to elicit meaning, gain understanding, and develop empirical knowledge [12]. We selected this approach because it allowed for an in-depth examination of how research at a specific institution addressed SDG 3 targets during a defined period, providing insights into the dynamics of pandemic response and sustainable development priorities.

For our analytical framework, we adapted the WHERETO model developed by Wiggins and McTighe [13] and integrated it with principles from Bloom’s Taxonomy [14]. While the WHERETO framework was originally designed for educational planning, we found its structured approach to backward design particularly valuable for analyzing how research aligns with predetermined sustainability targets. Table 1 presents our adaptation of this framework for research analysis. While the WHERETO framework provides a structure for analyzing research alignment with the SDG targets, understanding how this translates into health system improvements requires additional theoretical grounding. The Knowledge-to-Action (KTA) framework developed by Graham and colleagues [15] offers a complementary lens that bridges the gap between research generation and implementation. The KTA framework’s dual focus on knowledge creation (inquiry, synthesis, and tools) and an action cycle (problem identification, adaptation, implementation, and sustainability) aligns strategically with WHERETO’s backward design approach. Together these frameworks enable examinations of both what research is conducted (WHERETO) and how effectively it moves from academic inquiry to health system impact (KTA). This integrated approach is particularly relevant for understanding university research responses during health crises where rapid knowledge generation must translate quickly into actionable interventions.

We selected this analytical framework over alternatives for several reasons. First, its backward design approach aligned with our objective of examining how existing research addresses predetermined SDG targets, rather than allowing categories to emerge inductively from the data. Second, the framework facilitated a systematic evaluation of both the current research status and gaps. Third, its emphasis on stakeholder enablement (E) and reflection (R) provided a valuable lens for analyzing how research priorities might influence health strategies beyond academia. Finally, the framework’s organization component (O) offered an effective structure for translating findings into actionable recommendations.

### 2.2. Data Collection and Sampling

We employed a comprehensive census approach rather than selective sampling to capture the complete research landscape during the pandemic period. Our inclusion criteria were as follows: (1) research studies registered through the ethics committee approval process at UWC, (2) studies originating from health-related faculties (Faculty of Community and Health Sciences, Faculty of Natural Sciences, and Faculty of Dentistry), (3) registration dates between January 2020 and December 2022, and (4) the availability of research titles and abstracts in the ethics database. We excluded studies that were withdrawn before ethics approval and those with incomplete registration information.

This census approach was deliberate to avoid selection bias that might occur with purposive or stratified sampling methods. By including all registered studies rather than a sample, we aimed to provide a comprehensive picture of institutional research priorities during the pandemic period. The ethics committee registration point was selected as our sampling frame because it represents the formal commitment to conduct research and reflects institutional research priorities at the planning stage, thereby clarifying how studies were selected from the complete population of registered research.

Our sample included all research titles registered in the ethics projects database from the Faculty of Community and Health Sciences, Faculty of Natural Sciences, and Faculty of Dentistry at UWC between January 2020 and December 2022. We chose this comprehensive approach over selective sampling to ensure we captured the complete landscape of health research during the pandemic period. In total, we analyzed 889 research titles (300 from 2020, 313 from 2021, and 276 from 2022). The time period (2020–2022) was deliberately selected to capture research priorities during different phases of the COVID-19 pandemic, from the initial emergency response through adaptation and early recovery. This timing allowed us to observe how research agendas shifted in response to evolving pandemic conditions while maintaining focus on pre-existing health priorities.

### 2.3. Detailed Exclusion Criteria

To ensure methodological rigor and maintain data quality, we implemented the following specific exclusion criteria in our comprehensive census of health research at UWC:Studies withdrawn before ethics approval: We excluded research projects that were submitted to the ethics committee but withdrawn by the investigators prior to receiving formal approval. This was done to ensure that only committed research initiatives were included in our analysis.Incomplete registration information: Studies lacking complete research titles or abstracts in the ethics database were excluded as these components were essential for accurate SDG 3 target classification using our coding framework [13].Non-health-related research: Studies from faculties outside our defined health-related scope (excluding Faculty of Community and Health Sciences, Faculty of Natural Sciences, and Faculty of Dentistry) were excluded to maintain focus on health research contributions to SDG 3.Duplicate registrations: Multiple registrations of the same research project were identified, and duplicate entries were excluded to prevent inflated research counts.Conditional approvals without completion: Studies that received conditional ethics approval but did not meet final requirements for full approval were excluded, ensuring that only fully approved research was analyzed.Protocol registrations without study approval: Preliminary protocol submissions that did not advance to full study registration were excluded to focus on research projects with institutional commitment.

The application of the exclusion criteria resulted in a final sample of 889 research titles from an initial pool of 945 registered submissions, which represents a 5.9% exclusion rate. Most exclusions were due to studies that were withdrawn (3.2%) and incomplete registration information (2.1%), with the remaining exclusions spread across other criteria categories.

### 2.4. Coding and Analysis Procedure

We developed a systematic coding framework grounded in the official UN descriptions of each SDG 3 target and its associated indicators. This framework included detailed definitions, keywords, and contextual indicators for each SDG 3 target (see Appendix A). The coding process involved several steps:Initial codebook development: We created a comprehensive coding guide containing definitions, keywords, and contextual indicators for each SDG 3 target.Pilot coding and reliability testing: Two researchers independently coded the same subset of 100 randomly selected titles (approximately 11% of the total sample) to establish inter-rater reliability using Cohen’s kappa coefficient (κ = 0.82, indicating substantial agreement).Codebook refinement: Based on discrepancies identified during reliability testing, we refined our coding guidelines through discussion and consensus before proceeding with the full dataset.Full dataset coding: Each research title was assigned to the most relevant SDG 3 target using our coding framework. For ambiguous titles that potentially related to multiple targets, we applied a hierarchical decision tree that prioritized the primary focus of the research.Abstract consultation: When titles alone provided insufficient information for classification, we consulted project abstracts from the ethics database to ensure accurate categorization.Consensus meetings: Regular meetings were held by the research team to discuss uncertain cases and ensure consistent application of the coding framework.Temporal analysis: After completing the initial coding, we analyzed changes in the distribution of research titles across SDG 3 targets over the three-year period to identify trends related to pandemic phases.

The coding process paid particular attention to contextualizing research within South Africa’s health priorities as established by the National Health Research Committee [16] and the National Department of Health’s Digital Health Strategy [17]. This contextualization allowed us to assess how university research aligned with national health priorities while responding to the immediate challenges posed by the COVID-19 pandemic.

### 2.5. COVID-19 Context During Data Collection

Our data collection period (2020–2022) coincided directly with the global COVID-19 pandemic, which significantly influenced the research landscape at UWC and other institutions worldwide. This timing provided a unique opportunity to observe how research priorities shifted in response to an unprecedented global health crisis.

The pandemic created numerous challenges for conducting research, including disrupted fieldwork, limited access to communities, restricted laboratory operations, and redirected funding streams. It also stimulated new research questions, methodological innovations, and collaborative approaches. Our document analysis approach allowed us to capture these dynamics by examining titles registered during different phases of the pandemic response.

We paid particular attention to whether certain SDG 3 targets received heightened focus during the pandemic (such as Target 3.d on emergency preparedness) and whether traditional research priorities in the Western Cape context (HIV/AIDS, TB, and NCDs) maintained their prominence despite competing demands on the health system. This pandemic context provides an important lens for the interpretation of findings on how university research aligns with and responds to evolving health priorities during a global crisis.

Our analysis employed both frameworks synergistically: WHERETO’s components guided our systematic categorization of research against SDG 3 targets, while KTA principles informed our interpretation of how research priorities shifted in response to pandemic demands. Specifically, we examined evidence of knowledge adaptation (how existing research pivoted to address COVID-19), barrier identification (gaps in research coverage), and sustainability considerations (whether pandemic-responsive research-maintained momentum over time). This dual-framework approach enabled us to assess not only research quantity and focus areas but also indicators of knowledge translation potential, such as stakeholder relevance, local adaptation, and implementation readiness.

## 3. Results

Analysis of 889 research titles from 2020 to 2022 revealed that research at UWC varied significantly in how it addressed different SDG 3 targets. Table 2 presents the distribution of research titles across all 13 SDG 3 targets for each year of the study period.

The four most researched targets across the entire study period were as follows:Target 3.4 (NCDs and mental health): 445 titles (50.1% of total);Target 3.3 (infectious diseases): 130 titles (14.6% of total);Target 3.b (medicine and vaccine development): 125 titles (14.1% of total);Target 3.8 (universal health coverage): 111 titles (12.5% of total).

Conversely, the least researched targets were as follows:Target 3.6 (road injuries and deaths): 1 title (0.1% of total);Target 3.9 (environmental health): 4 titles (0.5% of total);Target 3.a (tobacco control): 11 titles (1.2% of total);Target 3.1 (maternal mortality): 14 titles (1.6% of total).

### 3.1. Pandemic-Related Research Trends

Analysis of the data reveals notable shifts in research priorities during different phases of the pandemic. While Target 3.4 (non-communicable diseases and mental health) consistently had the highest number of research titles across all three years, we observed significant fluctuations in other targets that correlate with pandemic response phases as indicated in Figure 1:

Emergency Preparedness (Target 3.d): Research focused on emergency preparedness surged from 0 titles in 2020 to 27 in 2021 and then slightly decreased to 22 in 2022. This pattern reflects the transition from initial crisis response to system strengthening for future preparedness. The emergence of emergency preparedness exemplifies rapid knowledge adaptation, which is a key principle of KTA [15].

Health Workforce (Target 3.c): Research on health worker density and distribution increased dramatically from 0 in 2020 to 80 in 2021, before declining to 21 in 2022. This trend likely indicates growing concerns about healthcare workforce capacity, resilience, and protection during peak pandemic periods. This increase in health workforce research demonstrates successful problem identification and knowledge adaptation phases in the KTA cycle [18,19].

Medicine and Vaccine Development (Target 3.b): We observed a consistent increase in research related to medicine and vaccine development, growing from 15 titles in 2020 to 50 in 2021 and further to 60 in 2022. This trend mirrors the global prioritization of pharmaceutical interventions for pandemic control.

Infectious Diseases (Target 3.3): Research on infectious diseases more than doubled from 32 titles in 2020 to 75 in 2021 before decreasing to 23 in 2022. This pattern likely reflects the initial focus on understanding COVID-19 transmission and its interaction with existing infectious disease burdens in South Africa, particularly HIV and tuberculosis.

Universal Health Coverage (Target 3.8): Research on health system access and coverage decreased from 63 titles in 2020 to 19 in 2021 before slightly increasing to 29 in 2022. This pattern may reflect shifting priorities during acute crisis phases and the challenges of conducting health systems research during pandemic restrictions.

These trends suggest that while researchers at UWC maintained focus on the region’s endemic health challenges (particularly NCDs), they adaptively responded to emerging priorities related to the pandemic. This demonstrates the university’s capacity to contribute to both immediate crisis responses and longer-term Sustainable Development Goals.

### 3.2. Alignment with Regional Health Priorities

According to the National Department of Health’s Digital Health Strategy (2019–2024) [17] and the National Health Research Committee priorities [16], the highest health priorities in South Africa include HIV/AIDS, tuberculosis, and non-communicable diseases. These health issues are particularly prevalent in disadvantaged communities within the Western Cape, where socioeconomic factors such as limited healthcare access, income inequality, and education disparities exacerbate health challenges.

Our analysis reveals a strong alignment between these national priorities and research emphasis at UWC.

However, we identified potential gaps in alignment with regional health priorities, particularly regarding the following:Injury prevention: Despite injuries forming part of South Africa’s quadruple burden of disease, Target 3.6 (road injuries and deaths) received minimal research attention.Environmental health determinants: Target 3.9 (environmental health hazards) received limited research focus despite growing evidence linking environmental factors to health outcomes in disadvantaged communities.Maternal and child health: Targets 3.1 (maternal mortality) and 3.2 (child mortality) received relatively little research attention compared to their significance in the South African context.

The heat map below illustrates the intensity of research focus across different SDG 3 targets throughout the pandemic period, revealing both consistent priorities and adaptive shifts in research attention.

This heat map visualization in the Figure 2 transforms the numerical data from Table 2 into a visual pattern that makes research intensity trends more apparent to readers. The actual study numbers for each cell are provided, derived directly from the research title counts presented in our results.

These findings suggest opportunities for strengthening research alignment with comprehensive regional health priorities while maintaining the adaptive capacity demonstrated during the pandemic response. This adaptive capacity aligns with global trends observed across universities during the pandemic, where institutions demonstrated both agility in crisis response and challenges in maintaining long-term research strategies [2,3].

The network diagram in Figure 3 below shows potential interconnections between SDG 3 targets based on research themes and shared methodological approaches identified in our analysis.

## 4. Discussion

The COVID-19 pandemic fundamentally challenged universities to balance immediate crisis response with long-term health development goals [1,2]. Our findings reveal a complex story of institutional adaptation, where UWC researchers demonstrated remarkable agility in addressing emerging health priorities while maintaining focus on endemic challenges like NCDs and infectious diseases. However, beneath these research shifts lie deeper questions about how academic institutions translate knowledge into lasting health system improvements [15,20]. This discussion examines what our integrated framework analysis reveals about university research’s evolving role in advancing sustainable development during and beyond health crises [3,4].

### 4.1. The Balancing Act: Crisis Response Meets Long-Term Development

UWC’s research response exemplifies the delicate balance universities must strike during health emergencies [2,4]. The institution demonstrated remarkable agility through dramatic research pivots; emergency preparedness research surged from 0 studies in 2020 to 27 in 2021, while health workforce research expanded from zero to 80 studies. From a knowledge translation perspective, this rapid scaling represents the successful problem identification and knowledge creation phases of the KTA cycle [15], where researchers quickly recognized critical knowledge gaps and mobilized institutional resources to address them.

Yet this agility coexisted with strategic continuity in core areas. Research on non-communicable diseases and mental health (Target 3.4) maintained its dominant position across all three years, representing 50.1% of total research, a testament to institutional commitment to addressing South Africa’s established health priorities [21,22]. This pattern reflects successful local knowledge adaptation, a key KTA principle [15,20], where researchers effectively balanced pandemic demands with contextually relevant health challenges that predated COVID-19.

The consistency in NCD research, alongside the surge in emergency preparedness, demonstrates institutional sophistication in resource allocation [23]. Rather than abandoning long-term development goals, UWC researchers expanded their portfolio to encompass both immediate crisis needs and endemic health challenges. This dual focus aligns with the WHERETO framework’s systematic target coverage while addressing the KTA cycle’s emphasis on sustained stakeholder engagement [20,24].

### 4.2. Knowledge Translation Successes and Barriers

The research adaptation patterns reveal both remarkable successes and critical barriers in knowledge translation pathways [20,24]. The rapid emergence of medicine and vaccine development research, growing consistently from 15 to 60 studies, represents sustained knowledge creation that maintained momentum throughout the pandemic period. This upward trajectory mirrors global pharmaceutical intervention priorities [23] and demonstrates how certain research areas can achieve both rapid adaptation and sustainability when supported by adequate resources and clear stakeholder alignment [15].

However, the KTA framework lens reveals concerning implementation challenges [25,26]. Emergency preparedness research declined from 27 to 22 studies in 2022, while health workforce research dropped dramatically from 80 to 21 studies. These patterns suggest that while universities excel at rapid knowledge adaptation during crisis periods, they struggle with sustainability planning and implementation and monitoring critical KTA cycle components to ensure research impacts beyond immediate emergency responses [20,24].

The research gaps in maternal health (1.6% of studies), environmental health (0.5%), and injury prevention (0.1%) represent more than simple research oversight. Applying KTA principles [15,20], these gaps indicate systemic barriers in stakeholder engagement and barrier assessment phases, where academic capacity may not adequately connect with community-identified priorities [27,28]. The near-absence of road injury research is particularly significant given that injury-related conditions form part of South Africa’s quadruple burden of disease [21,22], suggesting insufficient problem identification in the knowledge creation phase.

These translation barriers reflect what can be understood as implementation challenges in the KTA cycle [15], where traditional academic strengths in pharmaceutical and clinical research may not align with the complex stakeholder engagement required for addressing social determinants of health [27]. The concentration of research in NCDs while neglecting injury prevention and environmental health indicates that current research prioritization mechanisms may inadequately reflect the full spectrum of health challenges facing disadvantaged communities in the Western Cape [16,17].

### 4.3. The Deeper Questions: From Knowledge Creation to Health System Impact

Our analysis reveals fundamental tensions in how universities contribute to sustainable development during health crises [1,29]. The uneven distribution of research across SDG 3 targets exposes critical gaps in knowledge translation that require systematic policy intervention [24,25]. While UWC demonstrated an institutional capacity for rapid knowledge adaptation, the sustainability challenges observed in emergency preparedness and health workforce research indicate insufficient attention to the KTA cycle’s implementation monitoring and sustainability planning components [15,20].

The shift toward pharmaceutical research while health systems research declined from 63 to 19 studies in universal health coverage suggests a bias toward knowledge creation activities that align with traditional academic strengths rather than the more complex knowledge translation required for health systems transformation [30,31]. This pattern reflects barriers in the KTA cycle where research priorities may be driven more by institutional capacity and funding availability than by systematic assessment of knowledge translation potential and stakeholder needs [25,26].

From a knowledge translation perspective, current research-to-policy pathways appear episodic rather than systematic, lacking the dedicated knowledge translation units and implementation-monitoring systems that would support sustained research impacts [24,30]. These research gaps directly contradict Western Cape’s documented health priorities [16,17], suggesting insufficient communication and collaboration between health authorities and academic institutions during research planning phases [27,28].

### 4.4. The University’s Evolving Role: Toward Integrated Knowledge Translation

The integration of WHERETO and KTA frameworks reveals that an optimal pandemic response requires balancing rapid knowledge adaptation with systematic target coverage and explicit attention to knowledge translation pathways [15,20]. UWC’s experience demonstrates that universities can serve as catalysts for social change during crisis periods [4], but realizing this potential requires the stronger integration of knowledge creation with implementation planning [24,26].

Moving forward, the university’s evolving role demands comprehensive approaches that address both WHERETO’s systematic target coverage and KTA’s implementation requirements [15,32]. This includes annual collaboration between provincial health authorities and universities that connects knowledge creation with local adaptation needs [27], funding mechanisms requiring explicit knowledge translation plans [25], and sustainability mechanisms that maintain research focus beyond immediate crises while evaluating real-world health impact [29,30].

The integrated framework analysis suggests that effective university contributions to sustainable development during health crises require simultaneous attention to systematic target coverage and explicit knowledge translation planning, with particular emphasis on sustainability mechanisms that maintain research impact beyond immediate crisis response periods [20,24]. Future research should employ multi-institutional comparative approaches to distinguish between context-specific responses and generalizable patterns [2,3], ultimately contributing to more coordinated pandemic response strategies that strengthen both immediate crisis response and long-term sustainable development objectives [1,29].

This complex story of adaptation reveals universities’ critical yet challenging role in advancing global health security while pursuing sustainable development, a role that demands new frameworks to ensure that rapid knowledge creation translates into lasting improvements in health systems and community wellbeing [4,5].

### 4.5. Policy Impact Assessment and Transferability Framework

The methodology developed in this study offers significant potential for expansion to other academic environments through several key policy impact pathways. Our integrated WHERETO-KTA framework provides a replicable analytical structure that can inform evidence-based policy decisions at multiple levels.

At the institutional level, university leadership can utilize this framework to conduct annual research audits that assess alignment between institutional research portfolios and national health priorities. The systematic categorization of research against SDG 3 targets enables senior management to identify strategic gaps and redirect resources accordingly. For example, our findings revealing minimal research attention to injury prevention (0.1% of studies), despite its significance in South Africa’s quadruple burden of disease [8,9], could prompt targeted faculty recruitment or interdisciplinary collaboration initiatives.

At the policy level, health authorities can employ this methodology to evaluate whether academic research ecosystems adequately support national health strategies. The observed research gaps in maternal health (1.6% of studies) and environmental health (0.5% of studies) provide actionable intelligence for national research funding agencies to adjust priority areas and funding allocation mechanisms [16,17]. This evidence-based approach to research prioritization could strengthen the translation pathway between academic knowledge production and health system needs [15].

The methodology’s transferability is enhanced by its use of internationally standardized SDG 3 targets [1], enabling cross-institutional and cross-national comparisons. Universities in similar contexts, particularly those in low- and middle-income countries facing comparable health challenges, can adapt this framework by (1) mapping their research databases against SDG 3 targets using our coding framework, (2) contextualizing findings against their national health priorities [16], and (3) implementing the integrated WHERETO-KTA analysis to identify knowledge translation gaps [13,15].

Policy impact extends beyond academic institutions to inform national research strategy development. Our findings demonstrate how pandemic-responsive research can coexist with long-term development priorities [1,2], providing evidence for flexible research funding mechanisms that maintain core health research capacity while enabling rapid adaptation to emerging threats. This model could inform national research councils and international funding agencies in developing more responsive and sustainable research support systems [11].

## 5. Recommendations

Based on our analysis, we propose comprehensive recommendations that help health authorities to strengthen research contributions to Sustainable Development Goals.

### 5.1. Policy Recommendations for Health Authorities

Health departments must collaborate with universities to ensure research funding mechanisms support both immediate health system needs and long-term sustainable development objectives [16,17]. Establishing formal channels for regular dialog between health policymakers and university researchers will ensure research questions address policy-relevant priorities [24,25]. Health authorities should implement frameworks that enable the quick mobilization of university research capacity during health emergencies while maintaining ethical and quality standards [2,12]. Investment in implementation science initiatives that focus on translating evidence into practice is essential, particularly in areas where knowledge gaps impede effective health interventions [15,20]. Finally, facilitating partnerships between universities and community organizations will ensure research addresses grassroots health priorities and enables meaningful community participation in research design [27,28].

### 5.2. Limitations of the Study

While we acknowledge that our single-institution focus limits comparative analysis, this case study approach was deliberately chosen to provide a detailed framework that other universities can adapt to examine their pandemic research responses. We do not have access to comprehensive data from other institutions, but our methodology and analytical framework using the WHERETO model, combined with SDG 3 target mapping, offers a replicable approach that institutions can use to assess their research alignment with Sustainable Development Goals during crisis periods. This framework could enable other universities to conduct similar analyses of their pandemic research responses, ultimately contributing to a broader understanding of institutional adaptation patterns across different contexts and potentially informing the development of more coordinated pandemic response strategies in higher education

The study’s design, based on titles and abstracts, enabled efficient categorization, but it may not fully reflect the interdisciplinary nature or comprehensive scope of each research project. Future studies should consider in-depth content analysis of full proposals or published outputs to more accurately map contributions to multiple SDG targets. This analysis must be interpreted within important methodological and contextual limitations. First, our focus on a single institution limits the generalizability of findings and prevents us from determining whether observed patterns represent institution-specific responses or broader trends in university research during the pandemic. The research priority shifts we observed may reflect UWC’s particular resource constraints, researcher expertise, and community context rather than optimal or standardized responses to the pandemic.

A critical limitation of this study is the absence of comparative analysis with other universities, which would have enabled the triangulation of results to distinguish between pandemic-driven responses and institution-specific contexts. Future studies should employ multi-institutional comparative frameworks to determine whether the patterns we observed represent (a) generalizable responses to pandemic pressures, (b) context-specific adaptations unique to UWC’s institutional environment, or (c) optimal resource allocation strategies that other institutions might emulate. Such comparative analysis would better explain whether research shifts were primarily responses to pandemic-generated needs or reflections of UWC’s particular institutional capacity and constraints.

Second, our methodology captures research intentions at the planning stage (ethics approval) rather than completed research outcomes or the actual impact on health systems or communities. This approach provides valuable insights into institutional priorities but cannot assess whether the observed research addressed community needs, contributed to pandemic response effectiveness, or resulted in meaningful health improvements.

## 6. Conclusions

Moving forward, post-pandemic research strategies must emphasize comprehensive, equitable, and contextually informed priorities. Strengthening inter-institutional networks and investing in implementation science can ensure that research more directly informs health system resilience and community wellbeing.

The adaptive shifts in research priorities during the pandemic demonstrate UWC’s capacity to respond to emerging health threats, though these changes appear to reflect individual researcher initiatives and available funding opportunities rather than a coordinated institutional strategy. The rapid emergence of research on emergency preparedness and health workforce issues, followed by subsequent declines, suggests that pandemic-responsive research may represent temporary adaptations rather than sustainable reorientations of research capacity.

Our analysis reveals important knowledge gaps in SDG 3 coverage, particularly in maternal and child health, environmental health determinants, and injury prevention. However, it is crucial to note that our methodology captures research planning intentions rather than assessing actual health impacts or outcomes. We cannot conclude that these research gaps directly resulted in adverse health outcomes or that increased research in certain areas led to improved health indicators. The observed gaps represent differences in research attention rather than evidence of health system failures or community health impacts.

The concentration of research on specific SDG 3 targets reflects both regional health priorities and the immediate demands of pandemic response but may also indicate institutional path dependence and researcher expertise clustering specific to UWC’s context. The heterogeneous nature of these research responses may reflect institutional path dependencies and resource constraints rather than evidence-based pandemic response strategies, highlighting the need for comparative institutional analysis to distinguish between optimal adaptations and circumstantial responses. Understanding the mechanisms driving these research shifts, whether institutional mandates, funding streams, or researcher expertise, is essential for developing standardized pandemic response frameworks that could guide other universities in similar contexts. Without comparative analysis across multiple institutions, we cannot determine whether these patterns represent optimal responses to pandemic challenges or institution-specific adaptations that may not be applicable elsewhere.

Future research should employ multi-institutional comparative approaches to better distinguish between context-specific responses and generalizable patterns in university research adaptation during health crises. Such studies would enhance our understanding of how to build more resilient and responsive research systems that can address both immediate health emergencies and longer-term sustainable development objectives. Additionally, longitudinal studies tracking research outcomes and health system impacts would provide crucial evidence about the effectiveness of different institutional response strategies.

## Figures and Tables

**Figure 1 ijerph-22-01057-f001:**
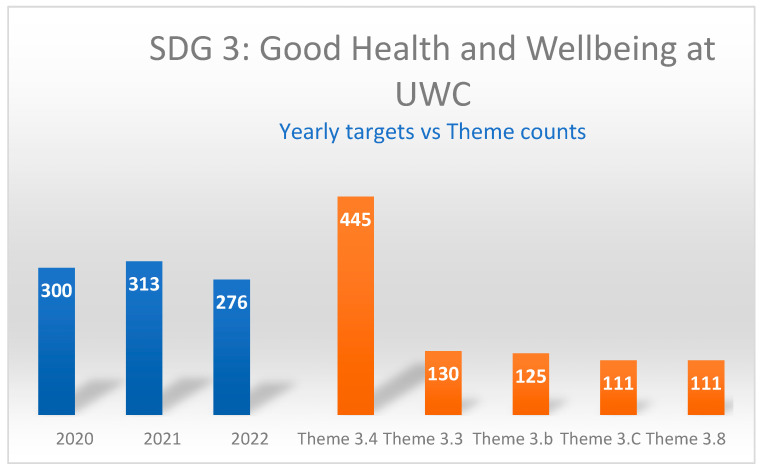
Distribution of research across top SDG 3 Targets (2020–2022). Source: Author’s analysis of data for the current study.

**Figure 2 ijerph-22-01057-f002:**
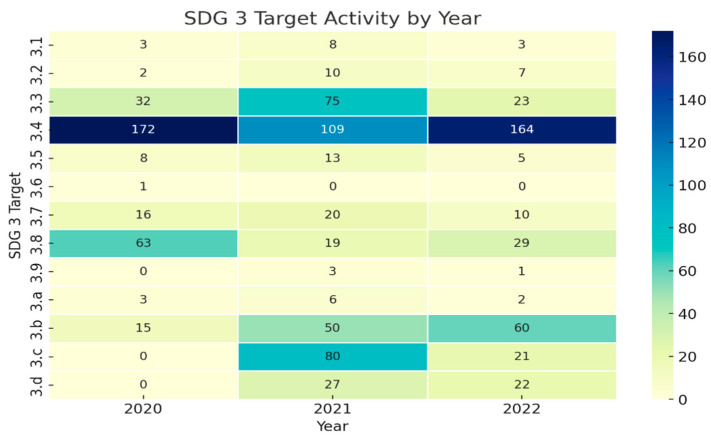
Heat map of research distribution across SDG 3 targets over time.

**Figure 3 ijerph-22-01057-f003:**
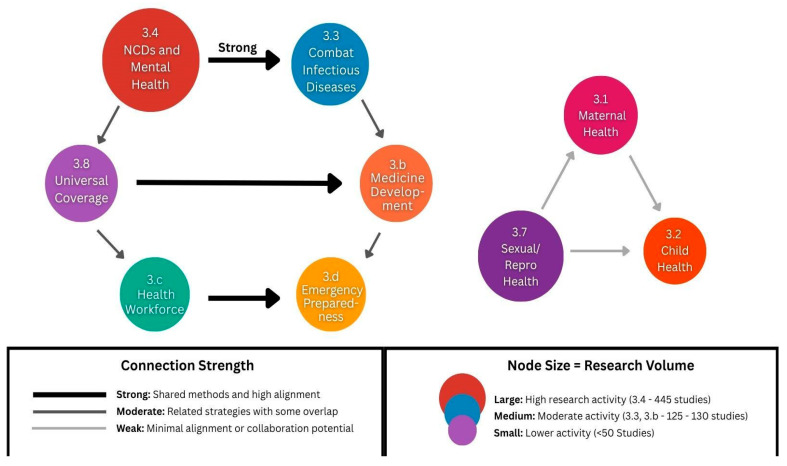
Network diagram of SDG target interconnections. Strong connections (thick lines): research areas with overlapping methodologies; moderate connections (medium lines): complementary research approaches; weak connections (thin lines): limited research intersection.

**Table 1 ijerph-22-01057-t001:** Integrated WHERETO-KTA framework for research analysis.

WHERETO Component	Original Definition in Education	Adaptation for Research Analysis	KTA Framework Alignment	Research Application
W (Where/Why)	Where and why is a way of thinking purposefully about curriculum planning?	Where and why is a way of purposefully thinking about research conducted?	Knowledge Inquiry Phase: Systematic investigation and problem identification	Analyzing research gaps and institutional priorities that drive SDG 3 target selection
H (Hook/Hold)	To hook and hold is to make meaning of big ideas	Hook and hold the big ideas and themes in research	Knowledge Synthesis: Capturing compelling research themes that engage stakeholders	Identifying research clusters around major health challenges (NCDs and infectious diseases)
E (Explore/Enable)	Explore, experience, enable, and equip	Explore how research is being used to enable various stakeholders	Knowledge Adaptation: Tailoring research to local contexts and stakeholder needs	Assessing how research addresses specific community health priorities and policy needs
R (Reflect)	Reflect, rethink, and revise	Reflection on evidence to identify priorities and purposes	Barrier Assessment: Identifying obstacles to knowledge translation and research impact	Evaluating gaps between research focus and health system implementation requirements
E (Evaluate)	Evaluate work and progress	Evaluation of research projects and progress toward addressing indicators	Implementation Monitoring: Tracking research contributions to health outcomes	Measuring research alignment with SDG 3 targets and indicators over time
T (Tailor)	Tailor and personalize	Tailor to and personalize for the institutional and regional context	Local Knowledge Adaptation: Contextualizing evidence for specific settings	Adapting research priorities to South African health challenges and Western Cape needs
O (Organize)	Organize for optimal effectiveness	Organize findings to identify gaps and future research directions	Sustainability Planning: Ensuring continued knowledge translation and impact	Structuring research portfolios for sustained contribution to SDG 3 achievement

Source: Adapted from [13].

**Table 2 ijerph-22-01057-t002:** Distribution of research titles across SDG 3 targets (2020–2022).

Target	Description	2020	2021	2022
3.1	Reduce maternal mortality	3	8	3
3.2	End preventable deaths under 5 years of age	2	10	7
3.3	Fight infectious/communicable diseases	32	75	23
3.4	Reduce NCDs and promote mental health	172	109	164
3.5	Prevent and treat substance abuse	8	13	5
3.6	Reduce road injuries and deaths	1	0	0
3.7	Universal access to sexual and reproductive healthcare	16	20	10
3.8	Achieve universal health coverage	63	19	29
3.9	Reduce deaths from environmental health hazards	0	3	1
3.a	Implement tobacco control	3	6	2
3.b	Support medicine and vaccine development	15	50	60
3.c	Health worker density and distribution	0	80	21
3.d	Emergency preparedness	0	27	22
Total		300	313	276

Source: Authors and co-authors’ work.

## Data Availability

The data presented in this study are available on request from the corresponding author due to institutional data sharing policies that require formal data sharing agreements.

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
