# Peer review of "Assessment of SDG 3 Research Priorities and COVID-19 Recovery Pathways: A Case Study from University of the Western Cape, South Africa"

_ijerph, 2025, doi:10.3390/ijerph22071057_

Round 1
Reviewer 1 Report
Comments and Suggestions for Authors
While the authors acknowledge their paper’s limited generalizability due to its single-institution focus (UWC), expanding the scope to include comparative data from other South African or African universities would significantly strengthen their conclusions. Given the method's adaptability, adding this comparative perspective seems feasible and valuable.
Another limitation is that the current methodology, focusing on titles and abstracts, might not fully capture each study’s contribution—especially those that are interdisciplinary and cover multiple SDG targets. Although this limitation is acknowledged, it might have been mitigated by a deeper content review of the selected papers.
While the authors reference the W.H.E.R.E.T.O model, the theoretical framework connecting university research to sustainable development implementation could be more robust. Including interviews or surveys with stakeholders (researchers, government officials, university administrators, health officials) would deepen understanding of the decision-making behind research prioritization and funding.
Moreover, the paper does not explore how funding priorities might have influenced research directions, an essential aspect, especially given pandemic-driven funding shifts. Addressing how financial streams affected research priorities could clarify whether research areas were selected strategically or were simply resource-dependent.
The analysis currently emphasizes research topics rather than their practical impacts, such as tangible health outcomes, policy influence, or community effects. Incorporating examples of impactful studies would enhance the paper's practical relevance.
Lastly, adding more sophisticated statistical analyses, like significance tests to quantify shifts over time, would strengthen the research findings' robustness.
To further improve this work, authors may like to consider building a stronger theoretical framework explicitly connecting research priorities with sustainable development and health systems, possibly using frameworks such as the Quadruple Helix Model or the Knowledge-to-Action Framework.
The document analysis may be supplemented with stakeholders' interviews or short surveys, capturing insights into the factors influencing research priorities during the pandemic.
Comparative data from other South African institutions would improve generalizability, even if only secondary data from published institutional reports.
A deeper discussion on policy implications emerging from the research, particularly around post-pandemic research priorities and how outputs translated into practical applications, should be conducted, as it is the main objective of the paper.
Funding streams and their impacts on research priorities can also be analyzed, possibly by reviewing calls for proposals, funding allocations, and resource availability shifts during the pandemic.
On the editorial aspects, some repetitions between the Results and Discussion sections were noted that can be amended, reviewing also abbreviations listed.
Though the point on limiting references to a certain time frame is noted, a few key references from 2023-2024 may strengthen the literature base, particularly concerning post-pandemic and recovery research prioritization.
Author Response
We uploading a document where we address all the reviewer comments

Reviewer 2 Report
Comments and Suggestions for Authors
Title: Suggestion to simplify the title or find another wording that is more understandable.
It should be aligned with the overall objective. When reading the project summary, the purpose of the study is perfectly understandable. However, when reading the title, it is incomprehensible. This is not a matter of the English language, but rather a lack of alignment between it and the purpose of the study.
Regarding the selection of studies, it is not clear what the selection criteria were for these articles, regardless of whether they were qualitative or selected from the ethics committee. It appears that all articles from the period were included, but then how were they selected?
Having explained the above, describing the studies that were carried out to a greater or lesser extent is not enough to understand the reasons for these changes and whether this could contribute to developing standard strategies. It does explain how this university and its researchers responded to the monument issue.
This may be due to the institutional context. Depending on local resources and each researcher's specialty, this could be the response to changes in research, but they do not respond to an event (pandemic) in a standardized way. A more in-depth analysis of this aspect of the results is needed.
While it can be understood that priorities changed as the pandemic progressed, no impact aspects were developed, so it is not clear why the conclusions address the persistence of the pandemic's impact on maternal and child health and other problems related to the determinants of environmental health. This is because the impact was not assessed. Please explain this section or improve the wording of this conclusion.
The gap is addressed in the same conclusion. In the methodological field, the gap refers to a significant difference in existing knowledge. Therefore, researchers should better account for the fact that certain research topics changed over the course of the pandemic. This could have been due to the contexts specific to the particular university. Perhaps a comparison with the evolution of research at other universities would have been appropriate to better capture these differences. A triangulation of these results could better explain whether this was more in response to the need generated by the pandemic or simply a specific context of the university where this research was conducted. Researchers should further discuss this aspect.

Author Response
We have uploaded one document with all the reviewer comments

Reviewer 3 Report
Comments and Suggestions for Authors
IJERPH-3654285: COVID-19 and SDG 3: Assessing Research Priorities and Recovery Pathways Through a Case Study from the University of the Western Cape, South Africa
I must commend the authors for this study. The study contributes to current discourse on the attainment of SDG 3 via transforming research findings into actionable policies. The authors adopted scientifically acceptable research method, results were well-presented and the entire manuscript was well-written. However, I observed some issues for the authors’ attention and corrections. They are as follows;
Topic: It is my opinion that the topic should be restructured to read thus; Assessment of SDG 3 Research Priorities and Covid-19 Recovery Pathways: A Case Study from University of the Western Cape, South Africa
Abstract:
Lines 26-27: This study offers critical insights into how university research priorities adapted during the pandemic. Shifted is more appropriate.
Introduction
- Authors should cross-check grammar to ensure consistency with the chosen American English. Examples;
Line10 shows progress towards while Lines 40, 358, 400 etc.: show progress toward SDG 3.
- Also, correct the following words/phrases as highlighted in the sentences below;
Line 41: Infectious disease burdens. Replace with Infectious diseases burden
Line 112: providing insights into the dynamics between pandemic response and sustainable development priorities. Replace with Dynamics of
Line 103: quality of life key aims of SDG 3. Replace with quality of life-key aims of SDG 3
Data Collection and Sampling
Lines 133-140: Justification for selecting the search period was stated twice. The time period (2020-2022) was deliberately selected to capture research priorities during different phases of the COVID-19 pandemic, from initial emergency response through adaptation and early recovery. 👈 This suffices as the justification for selecting the search period. So, the first reason should be deleted.
Coding and Analysis Procedure
Line 147: keywords, and contextual indicators for each SDG 3 target. Authors can provide the keywords, and indicators for each target as appendix.
Line 151: reliability testing.
Include the tool used for reliability test and the result.
Lines 185-186: This pandemic context provides an important lens through which to interpret findings……. Replace with for interpretation of findings
Author Response
We have uploaded one document with all the comments

Round 2
Reviewer 1 Report
Comments and Suggestions for Authors
This version shows a substantial improvement. However, please note that the paper still suffers from a lack of possible assessment of policy impact that may help understanding how and why the methodology can be expanded to other academic environments; citation format can be better standardized (as well as Reference format); recommendations are still insufficient as you may like to propose actions to university leadership, or elaborate policy recommendations to health authorities, or finally to propose a framework for sustainable research planning; a heat map showing research distribution across SDG targets over time + a network diagram showing interconnections between targets and trend analysis graphs may improve the readers' understanding of results. On sampling: exclusion criteria details may help.
Author Response
Thank you for your suggestions, please see the attachment.
